# Reproducibility Study of "Robust Fair Clustering: A Novel Fairness Attack and Defense Framework"

**Iason Skylitsis**[1,2], **Zheng Feng**[1,2], **Idries Nasim**[1,2], **Camille Niessink**[1,2]

**Reviewed on OpenReview:** `https://openreview.net/forum?id=H1hLNjwrGy`

## Abstract

Clustering algorithms play a pivotal role in various societal applications, where fairness is paramount to prevent adverse impacts on individuals. In this study, we revisit the robustness of fair clustering algorithms against adversarial attacks, affirming previous research findings that highlighted their susceptibility and the resilience of the Consensus Fair Clustering (CFC) model. Beyond reproducing these critical results, our work extends the original analysis by refining the codebase for enhanced experimentation, introducing additional metrics and datasets to deepen the evaluation of fairness and clustering quality, exploring novel attack strategies, including targeted attacks on new metrics and a combined approach for balance and entropy as well as an ablation study. These contributions validate the original claims about the vulnerability and resilience of fair clustering algorithms and broaden the research landscape by offering a more comprehensive toolkit for assessing adversarial robustness in fair clustering.

## 1 Introduction

Clustering algorithms play an important role in the analysis and interpretation of the vast amounts of data generated by automated data collection systems across various sectors (Rodriguez et al., 2019; Xu & Wunsch, 2005). While these algorithms are useful, they raise critical issues like privacy (Schneier, 2015; Song et al., 2022) and accountability in data use (Oppold & Herschel, 2020), while requiring transparency in their clustering performance (Selbst et al., 2019). In addition, clustering algorithms are widely used in a variety of societal applications, such as loan disbursement, medical treatment strategy, and recruitment (Tsai & Chen, 2010; Zitnik et al., 2019; Roy et al., 2020), highlighting the critical issue of fairness. Several fair clustering methods have been proposed to mitigate algorithmic bias and increase fairness (Backurs et al., 2019).

However, fair clustering has not yet been explored from an adversarial attack perspective. This aspect is crucial, as adversarial attacks aim to compromise the utility of fairness in these models (Chhabra et al., 2021; Mehrabi et al., 2021), potentially reversing the benefits of fair clustering. To tackle this gap, the authors of the reviewed study experimented with a *black-box* adversarial attack to assess the vulnerability of fair clustering algorithms. Their work also proposes a novel model, *Consensus Fair Clustering* (CFC), designed to be highly resilient to the proposed fairness attack (Chhabra et al., 2023).

This work aims to address the following goals:

- [**Reproducibility Study**] **Reproducing the results from the original paper:** We successfully reproduced the three main claims of the original paper. Firstly, our findings partially confirm that the *black-box* adversarial attack can reduce the fairness performance by perturbing a small percentage of protected group memberships. Secondly, we reproduced the claim that existing fair clustering algorithms lack robustness against adversarial influence. Lastly, our results validate the third claim that the CFC model exhibits high resilience against the proposed fairness attack.

---

[1]University of Amsterdam, Amsterdam, The Netherlands
[2]Equal contribution

- **[Extended Work] Improvement of the original code:** One of the contributions of our work involved transforming the code into a script format and integrating argument parsing for more streamlined experimentation. Moreover, we systematically structured repetitive code segments into distinct functions.

- **[Extended Work] Additional Metrics and Datasets:** To enhance our comprehension of clustering performance, particularly concerning fairness and clustering quality, this study integrated additional metrics. Moreover, we expanded the scope of our research by including two further datasets, enriching the depth and breadth of our analysis.

- **[Extended Work] Additional Attack Methods:** We experimented with the implementation of new attack methods, attacking some of the newly proposed metrics, as well as creating a dual-optimization problem for a combined Balance and Entropy attack.

- **[Extended Work] Ablation Study:** As part of our efforts to evaluate and improve the CFC model, we conducted an ablation study on focusing on the hyperparameters alpha ($\alpha$) and beta ($\beta$). We experimented with various values of $\alpha$ and $\beta$, and the impact of these hyperparameters on the model's performance was measured in terms of both clustering utility and fairness utility. We have included the results of our ablation study in Section 4.3 of our report and discussed the insights and implications of our findings.

## 2 Scope of reproducibility

This work investigates the reproducibility of the original paper by Chhabra et al. 2023, which addresses the problem of the vulnerability of fair clustering algorithms to adversarial attacks aimed at degrading fairness utility. The concerns about algorithmic fairness have led to growing interest in the literature on defining, evaluating, and improving fairness in Machine Learning algorithms (Pessach & Shmueli, 2022). The authors investigate the robustness of clustering models, namely Fair K-Center (KFC), Fair Spectral Clustering (FSC), and Scalable Fairlet Decomposition (SFD), to adversarial influence by using a *black-box* attack approach. Furthermore, they propose the Consensus Fair Clustering (CFC) model to achieve truly robust fair clustering. An explanation of the methodology, datasets, and metrics employed by the authors can be seen in Section 3.

The main claims that the paper made are as follows:

- **Claim 1:** The *black-box* adversarial attack outlined in the original paper is capable of degrading the fairness performance by perturbing a small percentage of protected group memberships in the examined fair clustering models: KFC, FSC, and SFD.

- **Claim 2:** KFC, FSC, and SFD demonstrate a lack of robustness to adversarial influence, exhibiting significant volatility in terms of fairness utility metrics such as Balance and Entropy.

- **Claim 3:** CFC exhibits high resilience against the proposed fairness attack, offering a robust solution for achieving fair clustering.

In addition to replicating the findings presented in the original paper, we conduct additional experiments to further evaluate the performance of the algorithms.

## 3 Methodology

The author's implementation of their code is publically available in their GitHub repository.[3] However, some implementation details for the Fair K-Center algorithm were missing, considering the attack used and the assigned budget[4] parameter for the optimization problem; therefore, we had to make some minor adjustments.

---

[3]https://github.com/anshuman23/CFC
[4]Number of calls to the objective function

### 3.1 Fairness Attack

In the original paper, the authors propose a novel black-box attack that aims to reduce the fairness utility of fair clustering algorithms by perturbing a small percentage of samples' protected group memberships. This is defined as a Fairness Attack. The threat model is defined as an adversary who can control a subset of the protected attributes, denoted as $G_A \subseteq G$, and observe the cluster outputs of a fair clustering algorithm $\mathcal{F}$, where $\mathcal{F}$ is unknown to the adversary. The goal is to find the optimal perturbations that minimize the fairness utility for the remaining samples, denoted as $G_D = G/G_A \subseteq G$, by perturbing $G_A$. This problem is formulated as a two-level hierarchical optimization problem (Anandalingam & Friesz, 1992), where the lower-level problem is the fair clustering problem and the upper-level problem is the attack problem. The attack optimization problem can be defined analytically using two mapping functions:

$\eta$ : Takes $G_A$ and $G_D$ as inputs and gives output $G = \eta(G_A, G_D)$, which is the combined group memberships for the entire dataset.

$\theta$ : Takes $G_D$ and an output cluster labeling from a clustering algorithm for the entire dataset as input, returns the cluster labels for only the subset of samples with group memberships in $G_D$.

Based on these notations, the optimization problem for the attacker is defined as: $\min_{G_A} \phi(\theta(O, G_D), G_D)$ s.t. $O = \mathcal{F}(X, K, \eta(G_A, G_D))$. The authors solve this problem using a zeroth-order optimization algorithm.

### 3.2 Model descriptions

The paper uses three state-of-the-art fair clustering algorithms, namely Fair K-Center (KFC) (Harb & Lam, 2020), Fair Spectral Clustering (FSC) (Kleindessner et al., 2019), and Scalable Fairlet Decomposition (SFD) (Backurs et al., 2019).

**Fair K-Center.** The algorithm aims to achieve fair clustering by using the k-center objective. The goal is to minimize the traditional clustering objective while ensuring that no protected group is unfairly over or under-represented within any cluster. This is achieved by partitioning a set of $N$ data points, each belonging to at least one of $l$ protected groups, into $k$ clusters.

**Fair Spectral Clustering.** The algorithm is a constrained version of Spectral Clustering (SC) and is a popular method for partitioning graph data with an incorporated fairness notion. This notion defines clustering as fair if each demographic group is proportionally represented in every cluster.

**Scalable Fairlet Decomposition.** The algorithm is a practical approximation of the fairlet decomposition algorithm, introduced in Chierichetti et al. (2018), that runs in nearly linear time.

In addition, the authors introduced a novel robust fair clustering algorithm, **Contrastive Fair Clustering (CFC)**, which aims to learn fair and transferable representations for clustering. It employs a contrastive learning framework to ensure that the learned representations are not only discriminative for clustering, but also fair with respect to protected attributes.

SFD influences CFC to ensure balanced micro-level representation. FSC informs CFC's aim for proportional demographic balance across clusters. KFC lays the foundation for CFC's fairness by modifying clustering objectives to protect against group bias. CFC integrates these insights through a contrastive learning framework, embedding fairness directly into its clustering process.

### 3.3 Datasets

The authors provided download links for the *MNIST-USPS* and *Office-31* (Saenko et al., 2010) datasets. Since the link for the cropped version of *Extended Yale face B* (*Yale*) (Lee et al., 2005) was not working, a torrent was used to download the *Yale* dataset. Additionally, the *Inverted UCI DIGITS* (*DIGITS*) (Xu et al., 1992) dataset was included in the author's repository. Furthermore, *Multi-task Facial Landmark* (*MTFL*) (Zhang et al., 2014) and uncropped *Yale* were utilized as additional datasets. Further information on the datasets, including characteristics, protected attributes, and descriptions, are provided in Table 1.

The *Office-31* dataset consists of three source domains: Amazon, Webcam, and DSLR, where we used DSLR and Webcam. In the case of *DIGITS*, we modified the images by inverting their pixel values.

| Dataset | Num. samples | Num. categories | Protected attribute | Description |
|---|---|---|---|---|
| *MNIST-USPS* | 3,800 | 10 | Sample source | Handwritten digits |
| *Office-31* | 1,293 | 31 | Domain source | Office objects |
| *DIGITS* | 3,594 | 10 | Source of image | Handwritten digits |
| *Yale* | 2,414 | 38 | Azimuth and elevation | Frontal-face |
| uncropped *Yale* | 2,414 | 38 | Azimuth and elevation | Full-body & Background |
| *MTFL* | 2,000 | 2 | Glasses usage | Face |

Table 1: Summary of the datasets used in our experimentation. We followed the code of the authors to select the number of samples used for *MNIST-USPS*, *Office-31*, *DIGITS*, and *Yale* (cropped and uncropped). For *MTFL*, we balanced the dataset by randomly selecting 2,000 images (each with and without glasses).

### 3.4 Hyperparameters

To reproduce the results of the paper, we primarily adhered to the same hyperparameters as those specified in the original study, whenever they were specified in the article. The hyperparameters used in this study are detailed in Appendix A.

### 3.5 Experimental setup and code

The reproduction of the results was based on the jupyter notebooks of the original authors. One of our contributions was restructuring the code into a script format, introducing argument parsing for easier experimentation, and organizing repeated code into functions. All experiments shown in this paper can easily be reproduced using our code, which is publicly available on GitHub.[5]

For the attack, we maintained consistency with the original study by using the same setup, including seeds, hyperparameters, and pre-computed labels. However, due to missing code for the KFC algorithm in the jupyter notebooks, we arbitrarily selected the budget for the optimization function. Additionally, as the type of attack used for the KFC algorithm was unspecified, we conducted experiments for both Balance and Entropy (Appendix B). Furthermore, since no code was provided to reproduce the figures, we attempted to approximate the partitions present in the baseline research using specific values ranging from 0 to 0.3.

For the defense, we employed three manual seeds (42, 46, and 48) for torch randomization. Additionally, the CFC model was trained using the Adam optimizer with a learning rate of 0.01.

The original paper uses four metrics along two dimensions, namely fairness utility and clustering utility, for performance evaluation. For clustering utility, we consider to use Normalized Mutual Information (NMI) (Strehl & Ghosh, 2002) and Unsupervised Accuracy (ACC) (Li & Ding, 2006). For fairness utility, we consider Balance (Chierichetti et al., 2018) and Entropy (Li et al., 2020). The definitions for these metrics are provided in Appendix C.

### 3.6 Computational requirements

We conducted all experiments on a computer cluster, utilizing an NVIDIA A100 GPU and an Intel Xeon Platinum 8360Y CPU, except for those related to the KFC algorithm, which were performed locally on an AMD Ryzen 7 4800H CPU with 16 threads. Since the `zoopt` package runs exclusively on the CPU, no GPU was necessary for the attack experiments. The total computational cost for running all experiments amounted to roughly 80 CPU hours and 130 GPU hours.

## 4 Results

### 4.1 Results reproducing original paper

As stated in Section 2, three claims were identified in the original paper, and we were able to partially reproduce the first claim and entirely reproduce the second and third claims. In this section, we elaborate

---

[5]https://github.com/iasonsky/FACT-2024

on our reproduction results: first, in section 4.1.1 and 4.1.2, we show the results of the proposed *black-box* and random attack (Claims 1 and 2). In section 4.1.3, we show the results of the defense (Claim 3).

### 4.1.1 Claim 1: The *black-box* adversarial attack results in substantial degradation of fairness performance in Fair K-Center (KFC), Fair Spectral Clustering (FSC), and Scalable Fairlet Decomposition (SFD) by perturbing a small % of protected group memberships. [Partially Reproduced]

To validate Claim 1, we compared the attack with the random attack and present the results when 15% group memberships are switched for *MNIST-USPS* and *Office-31* in Table 2 and for *Inverted UCI DIGITS* (*DIGITS*) and *Extended Yale face B* (*Yale*) in Appendix E, Table 9.

Contrasting with the findings of the original paper, our results showed a slight increase, rather than a significant reduction, in fairness utility for SFD in terms of Balance/Entropy on the MNIST-USPS dataset (+6.382%/+1.339%). Similarly, for FSC in terms of Entropy on the Office-31 dataset, we observed a modest increase (+2.390%). Interestingly, the random attack sometimes led to increased fairness utility, notably a +100.0% increase in FSC Balance on Office-31. However, for KFC on Office-31, Balance/Entropy metrics remained unchanged at -0.000%/-0.000% (as discussed in Section 4.2.1). Despite these individual variations, a consistent reduction in fairness was noted across the other datasets post-attack, indicating a partial reproduction of the original claim.

In Appendix D, Table 8, we present a detailed comparison of the *Change (%)* values between our study and the baseline research. This analysis highlights the disparities in the impact of adversarial attacks on fairness and clustering metrics, offering insight into the relative robustness of the algorithms examined.

| Algorithm | Metrics | MNIST-USPS | | | | | | |
| --- | --- | --- | --- | --- | --- | --- | --- | --- |
| | | Pre-Attack | Post-Attack | Change (%) | Match Original Findings | Random Attack | Change (%) | Match Original Findings |
| SFD | Balance | 0.282 ± 0.001 | 0.300 ± 0.001 | (+)**6.382** | | 0.330 ± 0.001 | (+)17.02 | |
| | Entropy | 3.063 ± 0.151 | 3.104 ± 0.001 | (+)**1.339** | | 3.147 ± 0.000 | (+)2.742 | |
| | NMI | 0.315 ± 0.000 | 0.358 ± 0.000 | (+)13.65 | | 0.346 ± 0.000 | (+)9.841 | |
| | ACC | 0.419 ± 0.000 | 0.473 ± 0.000 | (+)12.89 | | 0.456 ± 0.000 | (+)8.831 | |
| FSC | Balance | 0.000 ± 0.000 | 0.000 ± 0.000 | (-)100.0 | ✓ | 0.000 ± 0.000 | (-)100.0 | ✓ |
| | Entropy | 0.327 ± 0.000 | 0.241 ± 0.000 | (-)26.30 | ✓ | 0.301 ± 0.001 | (-)7.951 | ✓ |
| | NMI | 0.549 ± 0.000 | 0.543 ± 0.000 | (-)1.093 | ✓ | 0.538 ± 0.000 | (-)2.004 | ✓ |
| | ACC | 0.450 ± 0.000 | 0.454 ± 0.000 | (+)0.889 | ✓ | 0.443 ± 0.000 | (-)1.556 | ✓ |
| KFC | Balance | 0.557 ± 0.324 | 0.350 ± 0.299 | (-)37.16 | ✓ | 0.724 ± 0.117 | (+)30.20 | ✓ |
| | Entropy | 1.355 ± 0.374 | 1.202 ± 0.351 | (-)11.29 | ✓ | 1.417 ± 0.417 | (+)4.576 | ✓ |
| | NMI | 0.000 ± 0.000 | 0.000 ± 0.000 | (-)100.0 | ✓ | 0.000 ± 0.000 | (-)100.0 | ✓ |
| | ACC | 0.147 ± 0.000 | 0.146 ± 0.000 | (-)0.680 | ✓ | 0.145 ± 0.000 | (-)1.361 | ✓ |
| Algorithm | Metrics | Office-31 | | | | | | |
| | | Pre-Attack | Post-Attack | Change (%) | Match Original Findings | Random Attack | Change (%) | Match Original Findings |
| SFD | Balance | 0.546 ± 0.000 | 0.158 ± 0.000 | (-)71.06 | ✓ | 0.359 ± 0.120 | (-)34.25 | ✓ |
| | Entropy | 10.00 ± 0.000 | 9.783 ± 0.001 | (-)2.170 | ✓ | 9.903 ± 0.001 | (-)0.970 | ✓ |
| | NMI | 0.888 ± 0.000 | 0.861 ± 0.000 | (-)3.041 | ✓ | 0.860 ± 0.000 | (-)3.153 | ✓ |
| | ACC | 0.841 ± 0.000 | 0.765 ± 0.000 | (-)9.037 | ✓ | 0.769 ± 0.000 | (-)8.561 | ✓ |
| FSC | Balance | 0.000 ± 0.000 | 0.000 ± 0.000 | (-)100.0 | ✓ | 0.211 ± 0.211 | (+)**100.0** | ✓ |
| | Entropy | 9.164 ± 0.119 | 9.383 ± 0.301 | (+)**2.390** | ✓ | 9.628 ± 0.213 | (+)5.063 | ✓ |
| | NMI | 0.652 ± 0.000 | 0.682 ± 0.000 | (+)4.601 | ✓ | 0.685 ± 0.000 | (+)5.061 | ✓ |
| | ACC | 0.390 ± 0.000 | 0.438 ± 0.000 | (+)12.31 | ✓ | 0.436 ± 0.000 | (+)18.72 | ✓ |
| KFC | Balance | 0.971 ± 0.001 | 0.971 ± 0.001 | (-)**0.000** | | 0.971 ± 0.001 | (-)0.000 | |
| | Entropy | 0.401 ± 0.135 | 0.401 ± 0.135 | (-)**0.000** | | 0.401 ± 0.135 | (-)0.000 | |
| | NMI | 0.000 ± 0.000 | 0.000 ± 0.000 | (-)100.0 | | 0.000 ± 0.000 | (-)100.0 | |
| | ACC | 0.001 ± 0.000 | 0.001 ± 0.000 | (-)0.000 | | 0.001 ± 0.000 | (-)0.000 | |

Table 2: Results for pre-attack, post-attack (*black-box*), random attack, change between pre- and post-attack / random attack, when 15% group membership labels are switched for fair clustering algorithms SFD, FSC, and KFC and datasets *MNIST-USPS* and *Office-31*. Results show the impact on fairness utility (Balance and Entropy) and clustering utility (NMI and ACC). The checkmarks indicate that for the specified dataset and algorithm combination the performance matches the findings in the baseline research.

### 4.1.2 Claim 2: KFC, FSC, and SFD demonstrate a lack of robustness to adversarial influence. [Reproduced]

In order to validate Claim 2, we replicated the experiment conducted by the authors, showcasing the pre-attack and post-attack results for *MNIST-USPS* and *Office-31* datasets using both the *black-box* attack and random attack, as illustrated in Figure 1. The results for *DIGITS* and *Yale* are shown in Appendix E, Figure 3.

We found that the fairness attack consistently outperforms the random attack baseline across both Balance and Entropy fairness metrics. However, contrary to the original claim, our findings revealed an exception for Entropy on the FSC algorithm applied to the *Office-31* dataset (Figure 1, Row 4, Column 2).

Additionally, the random attack does not consistently lead to lower fairness metric values. For instance, we observed an increase in both Balance and Entropy on the FSC algorithm applied to the *Office-31* dataset (Figure 1, Row 4, Column 1-2), which is consistent with the original paper's findings.

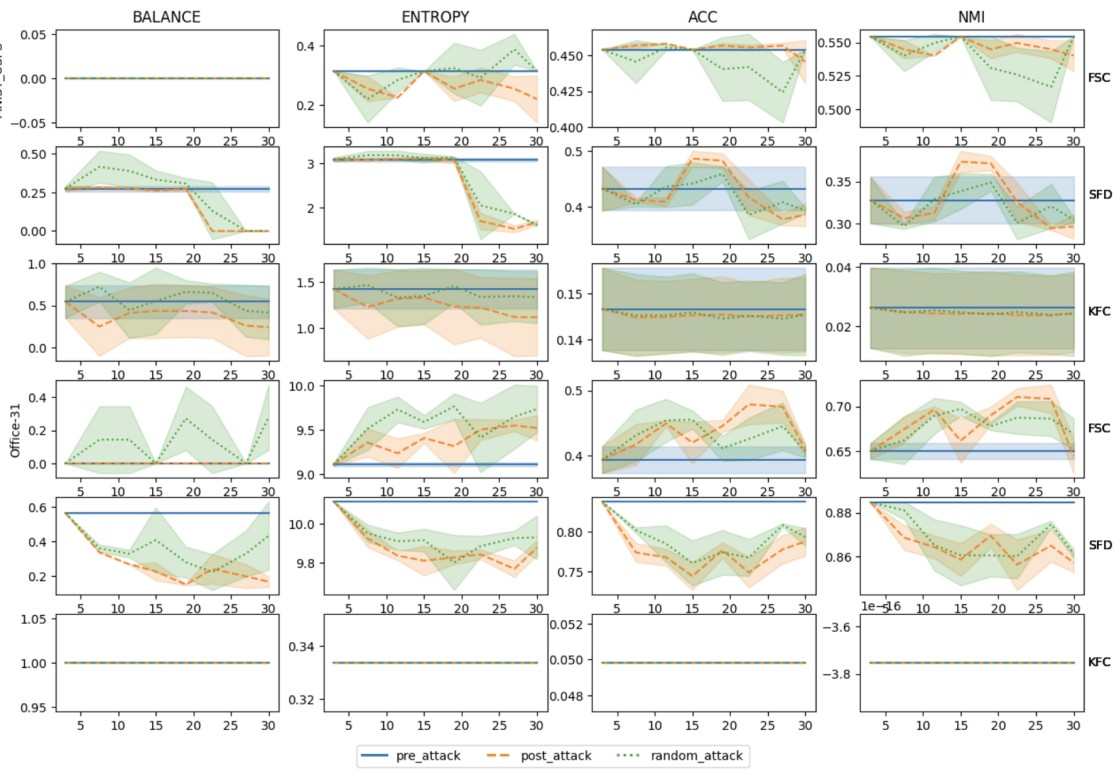

Figure 1: Pre-attack, post-attack (*black-box*) and random attack results on fairness utility (Balance and Entropy) and clustering utility (ACC and NMI) for *MNIST-USPS* and *Office-31* (x-axis: % of samples attacker can poison).

### 4.1.3 Claim 3: Contrastive Fair Clustering (CFC) exhibits high resilience against the proposed fairness attack. [Reproduced]

In order to validate Claim 3, we replicated the experiment conducted by the authors, showcasing the efficacy of the CFC algorithm in Table 3. This table provides a clear overview of the changes in fairness and clustering metrics before and after the application of the *black-box* adversarial attack, specifically when 15% of group membership labels are altered. It highlights the variations in fairness utility, measured by Balance and Entropy, and clustering utility, represented by NMI and ACC.

The analysis following the attack reveals that the CFC algorithm sustained its performance levels effectively. Notably, the fairness metrics demonstrated stability, with a minor exception observed in the Balance metric for the *MNIST-USPS* dataset, which decreased by 10.35%. Furthermore, clustering metrics either remained consistent or showed improvement post-attack. This improvement was particularly prominent in the NMI and ACC metrics for the *MNIST-USPS* dataset, which increased by 19.34% and 13.41%, respectively. In contrast, other fair clustering algorithms exhibited notable declines in fairness utility following the attack. Specifically, the FSC algorithm showed a significant reduction in the Balance and Entropy metrics for the *MNIST-USPS* dataset, dropping by 100.0% and 26.30%, respectively. Similarly, the SFD algorithm experienced a substantial decrease of 71.06% in the Balance metric for the *Office-31* dataset.

For an in-depth exploration of the CFC algorithm's resilience to adversarial challenges, readers are referred to Appendix G, Figure 4, which delves deeper into the robustness of the CFC model under adversarial conditions. Additionally, the findings related to the *DIGITS* and *Yale* datasets are detailed in Appendix F, Table 10, which also contrasts the percentage changes between our findings and those from the baseline research, further illustrating the comparative robustness of the CFC algorithm.

| Algorithm | Metric | MNIST-USPS | | | | Office-31 | | | |
|---|---|---|---|---|---|---|---|---|---|
| | | Pre-Attack | Post-Attack | Change (%) | Match Original Findings | Pre-Attack | Post-Attack | Change (%) | Match Original Findings |
| CFC | Balance | 0.470±0.041 | 0.421±0.027 | (-)10.35 | ✓ | 0.620±0.001 | 0.637±0.000 | (+)2.742 | ✓ |
| | Entropy | 2.622±0.128 | 2.689±0.126 | (+)2.555 | ✓ | 6.116±0.000 | 5.821±0.152 | (-)4.823 | ✓ |
| | NMI | 0.243±0.000 | 0.290±0.000 | (+)19.34 | ✓ | 0.693±0.000 | 0.681±0.000 | (-)1.732 | ✓ |
| | ACC | 0.358±0.000 | 0.406±0.000 | (+)13.41 | ✓ | 0.503±0.000 | 0.466±0.000 | (-)7.356 | ✓ |
| SFD | Balance | 0.282±0.001 | 0.300±0.001 | (+)6.382 | | 0.546±0.000 | 0.158±0.000 | (-)71.06 | ✓ |
| | Entropy | 3.063±0.151 | 3.104±0.001 | (+)1.339 | | 10.00±0.000 | 9.783±0.001 | (-)2.170 | ✓ |
| | NMI | 0.315±0.000 | 0.358±0.000 | (+)13.65 | | 0.888±0.000 | 0.861±0.000 | (-)3.041 | ✓ |
| | ACC | 0.419±0.000 | 0.473±0.000 | (+)12.89 | | 0.147±0.000 | 0.146±0.000 | (-)9.037 | ✓ |
| FSC | Balance | 0.000±0.000 | 0.000±0.000 | (-)100.0 | ✓ | 0.000±0.000 | 0.000±0.000 | (-)100.0 | ✓ |
| | Entropy | 0.327±0.000 | 0.241±0.001 | (-)26.30 | ✓ | 9.164±0.119 | 9.383±0.301 | (+)2.390 | ✓ |
| | NMI | 0.549±0.000 | 0.543±0.000 | (-)1.093 | ✓ | 0.652±0.000 | 0.682±0.000 | (+)4.601 | ✓ |
| | ACC | 0.450±0.000 | 0.454±0.000 | (+)0.889 | ✓ | 0.390±0.000 | 0.438±0.000 | (+)12.31 | ✓ |
| KFC | Balance | 0.557±0.324 | 0.350±0.299 | (-)37.16 | ✓ | 0.971±0.001 | 0.971±0.001 | (-)0.000 | |
| | Entropy | 1.355±0.374 | 1.202±0.351 | (-)11.29 | ✓ | 0.401±0.135 | 0.401±0.135 | (-)0.000 | |
| | NMI | 0.000±0.000 | 0.000±0.000 | (-)100.0 | ✓ | 0.000±0.000 | 0.000±0.000 | (-)100.0 | |
| | ACC | 0.147±0.000 | 0.146±0.000 | (-)0.680 | ✓ | 0.001±0.000 | 0.001±0.000 | (-)0.000 | |

Table 3: Results for pre-attack, post-attack (*black-box*), change between pre- and post-attack, when 15% group membership labels are switched for fair clustering algorithms CFC, SFD, FSC, and KFC and datasets *MNIST-USPS* and *Office-31*. Results show the impact on fairness utility (Balance and Entropy) and clustering utility (NMI and ACC). The checkmarks indicate that for the specified dataset and algorithm combination the performance matches the findings in the baseline research.

## 4.2 Results beyond original paper

### 4.2.1 Additional Metrics and Datasets

**Motivation:** In our study, we enhanced the original set of metrics - Balance, Entropy, NMI, and ACC - with additional metrics for a more comprehensive clustering analysis. First, the Adjusted Rand Index (ARI) complements ACC by evaluating clustering similarity without relying on label information. Next, the Silhouette Score measures cluster quality, while the Minimum Cluster Ratio evaluates group representation within clusters, a dimension overlooked by Balance. Metrics such as Cluster Distribution KL and Cluster Distribution Total Variation provide a deeper understanding of group distribution across clusters. The Silhouette Difference metric highlights disparities in clustering quality among groups. Additionally, separate calculations for Minority Cluster Distribution Entropy for Group A (sensitive attribute label = 0) and Group B (sensitive attribute label = 1) enrich our fairness assessment by analyzing the distributional homogeneity of each group. Furthermore, we included the *Multi-task Facial Landmark* (*MTFL*) and uncropped *Yale* datasets to broaden the scope of our analysis, adding complexity with diverse facial features and landmark annotations. This choice complements the existing *DIGITS* and cropped *Yale* datasets, enriching the study with real-world applicability and a more comprehensive assessment of clustering algorithm fairness and adaptability.

The results for the additional metrics on the *MNIST-USPS* and *Office-31* datasets are presented in Table 4, while results for the *DIGITS* and *Yale* datasets can be found in Appendix H, Table 11. To calculate the fairness metrics, we utilized the `holisticai` library while `scikit-learn` (Pedregosa et al., 2011) was used to assess the quality of clustering. The results for the two new datasets are shown in Table 5, where we see an increase in fairness utility for CFC in terms of Balance on the uncropped *Yale* dataset ((+)1.304%).

Our extended analysis revealed a unique pattern in the KFC algorithm, which consistently grouped all data points into a single cluster, an issue first identified through errors in calculating Silhouette Scores and Differences. This led to 'N/A' entries for these metrics in Table 4 and Appendix H, Table 11. Furthermore, we observed instances of infinite KL divergence, especially when Balance was 0, highlighting significant group distribution discrepancies across clusters. The remaining metrics, except for the Minimum Cluster Ratio, did not have significant variations pre- and post-attack.

In the uncropped *Yale* dataset, our findings highlighted the superior performance of the CFC algorithm, which exhibited a notable 130% increase in Balance post-attack. On the other hand for *MTFL* the results before and after showed no significant changes for all the tested algorithms.

| Algorithm | Metric | MNIST-USPS | | | Office-31 | | |
|---|---|---|---|---|---|---|---|
| | | Pre-Attack | Post-Attack | Random Attack | Pre-Attack | Post-Attack | Random Attack |
| SFD | Min. Cluster Ratio | $0.425 \pm 0.094$ | $0.500 \pm 0.022$ | $0.466 \pm 0.156$ | $0.269 \pm 0.015$ | $0.065 \pm 0.010$ | $0.138 \pm 0.065$ |
| | Cluster L1 | $0.276 \pm 0.077$ | $0.270 \pm 0.034$ | $0.276 \pm 0.122$ | $0.170 \pm 0.008$ | $0.180 \pm 0.006$ | $0.178 \pm 0.011$ |
| | Cluster KL | $0.294 \pm 0.134$ | $0.235 \pm 0.053$ | $0.269 \pm 0.300$ | $0.082 \pm 0.006$ | $0.100 \pm 0.005$ | $0.098 \pm 0.008$ |
| | Silhouette diff | $-0.015 \pm 0.008$ | $-0.020 \pm 0.009$ | $-0.019 \pm 0.006$ | $-0.006 \pm 0.001$ | $-0.008 \pm 0.002$ | $-0.008 \pm 0.002$ |
| | Entropy Group A | $2.264 \pm 0.006$ | $2.266 \pm 0.010$ | $2.173 \pm 0.290$ | $3.363 \pm 0.002$ | $3.292 \pm 0.014$ | $3.305 \pm 0.024$ |
| | Entropy Group B | $2.004 \pm 0.160$ | $2.070 \pm 0.069$ | $2.127 \pm 0.074$ | $3.353 \pm 0.005$ | $3.357 \pm 0.008$ | $3.354 \pm 0.010$ |
| | ARI | $0.201 \pm 0.037$ | $0.264 \pm 0.017$ | $0.248 \pm 0.046$ | $0.752 \pm 0.009$ | $0.687 \pm 0.022$ | $0.683 \pm 0.019$ |
| | Silhouette score | $0.021 \pm 0.011$ | $0.035 \pm 0.003$ | $0.039 \pm 0.011$ | $0.172 \pm 0.002$ | $0.158 \pm 0.005$ | $0.159 \pm 0.004$ |
| FSC | Min. Cluster Ratio | $0.000 \pm 0.000$ | $0.000 \pm 0.000$ | $0.000 \pm 0.000$ | $0.000 \pm 0.000$ | $0.000 \pm 0.000$ | $0.060 \pm 0.092$ |
| | Cluster L1 | $0.728 \pm 0.001$ | $0.846 \pm 0.080$ | $0.760 \pm 0.063$ | $0.112 \pm 0.010$ | $0.117 \pm 0.014$ | $0.113 \pm 0.015$ |
| | Cluster KL | $\infty^* \pm nan^*$ | $\infty^* \pm nan^*$ | $\infty^* \pm nan^*$ | $0.069 \pm 0.004$ | $0.064 \pm 0.008$ | $0.058 \pm 0.009$ |
| | Silhouette diff | $-0.069 \pm 0.002$ | $-0.068 \pm 0.003$ | $-0.070 \pm 0.002$ | $-0.009 \pm 0.002$ | $-0.004 \pm 0.008$ | $-0.007 \pm 0.007$ |
| | Entropy Group A | $1.150 \pm 0.002$ | $1.150 \pm 0.001$ | $1.151 \pm 0.003$ | $2.299 \pm 0.037$ | $2.489 \pm 0.091$ | $2.471 \pm 0.103$ |
| | Entropy Group B | $1.854 \pm 0.031$ | $1.862 \pm 0.031$ | $1.868 \pm 0.032$ | $2.385 \pm 0.047$ | $2.542 \pm 0.110$ | $2.569 \pm 0.094$ |
| | ARI | $0.259 \pm 0.010$ | $0.275 \pm 0.009$ | $0.260 \pm 0.017$ | $0.207 \pm 0.008$ | $0.223 \pm 0.033$ | $0.235 \pm 0.029$ |
| | Silhouette score | $0.036 \pm 0.000$ | $0.050 \pm 0.009$ | $0.040 \pm 0.008$ | $0.002 \pm 0.004$ | $0.021 \pm 0.013$ | $0.018 \pm 0.010$ |
| KFC | Min. Cluster Ratio | $0.603 \pm 0.341$ | $0.358 \pm 0.351$ | $0.696 \pm 0.279$ | $0.626 \pm 0.018$ | $0.626 \pm 0.018$ | $0.612 \pm 0.061$ |
| | Cluster L1 | $0.013 \pm 0.008$ | $0.018 \pm 0.010$ | $0.014 \pm 0.009$ | $0.000 \pm 0.000$ | $0.000 \pm 0.000$ | $0.000 \pm 0.000$ |
| | Cluster KL | $0.003 \pm 0.001$ | $0.005 \pm 0.002$ | $\infty^* \pm nan^*$ | $0.000 \pm 0.000$ | $0.000 \pm 0.000$ | $0.000 \pm 0.000$ |
| | Silhouette diff | $0.0262 \pm 0.023$ | $0.035 \pm 0.029$ | $0.022 \pm 0.026$ | N/A* | N/A* | N/A* |
| | Entropy Group A | $0.238 \pm 0.143$ | $0.201 \pm 0.132$ | $0.223 \pm 0.143$ | $0.007 \pm 0.015$ | $0.007 \pm 0.015$ | $0.006 \pm 0.012$ |
| | Entropy Group B | $0.275 \pm 0.164$ | $0.254 \pm 0.168$ | $0.258 \pm 0.170$ | $0.007 \pm 0.017$ | $0.007 \pm 0.017$ | $0.007 \pm 0.017$ |
| | ARI | $0.0 \pm 0.002$ | $0.0 \pm 0.001$ | $0.0 \pm 0.002$ | $0.000 \pm 0.000$ | $0.000 \pm 0.000$ | $0.000 \pm 0.000$ |
| | Silhouette score | $0.099 \pm 0.058$ | $0.113 \pm 0.058$ | $0.101 \pm 0.057$ | N/A* | N/A* | N/A* |

Table 4: Results for pre-attack, post-attack (*black-box*), and random attack, when 15% group membership labels are switched for fair clustering algorithms SFD, FSC, and KFC and datasets *MNIST-USPS* and *Office-31*. Results show the impact on additional metrics, where N/A corresponds to uniform clustering, $\infty$ to infinite values, and *nan* to undefined values.

### 4.2.2 Additional Attack Methods

**Motivation:** We introduced new and combined attacks targeting various evaluation metrics to test the model's robustness. This approach aimed to uncover potential vulnerabilities, guiding improvements for enhanced model generalization and defense.

We experimented with various attack strategies, focusing on the *Office-31* dataset using the SFD algorithm. Initially, our focus was to challenge one of the recently introduced fairness metrics, specifically the Minimum Cluster Ratio. This attempt yielded identical results to those of the balance attack, suggesting a complete correlation between these two metrics. Subsequently, we explored a dual-attack approach targeting both Balance and Entropy, using a different optimization toolbox, namely `Nevergrad` (Rapin & Teytaud, 2018), which uses a meta-optimizer named `NGOpt`.

In our study, we conducted a grid search across various weights for Balance and Entropy [1, 5, 10] and different budget levels [20, 40, 60], utilizing three seeds [42, 123, 456]. This process identified an optimal configuration with weights of 5 and 1 for Balance and Entropy, respectively, and a budget of 20. However, the improvements observed in the attack performance with this configuration, as compared to the originally proposed attack, were marginal. Detailed results of these comparisons are presented in Appendix I, Table 12. The new attacks on the Minimum Cluster Ratio and the combined attack were also tested with the CFC defense algorithm. The CFC algorithm demonstrated robustness in countering these attacks. The details of the results are shown in Appendix J, Table 13.

### 4.3 Ablation Study of Alpha and Beta Hyperparameters in the Consensus Fair Clustering Model

**Motivation:** The study mainly focused on two key hyperparameters, alpha ($\alpha$) and beta ($\beta$), which were tuned to optimize for training loss. Alpha controls the ratio of fair clustering loss, and beta controls the ratio of structural preservation loss. The ablation study was conducted to understand the effects of these two hyperparameters on the performance of the CFC model.

We trained the Consensus Fair Clustering (CFC) model on the Cora dataset (McCallum et al., 2000) This dataset consists of 2708 scientific publications classified into seven classes and was balanced by randomly sampling 1000 papers and using the binary feature 'w_1177' as the sensitive attribute. We evaluated the

CFC model's clustering and fairness performance under different hyperparameter settings. Some of the hyperparameters for the CFC model were kept constant, such as the number of basic partitions (r = 100), the temperature parameter in the contrastive loss ($\tau = 1$), dropout in hidden layers (0.6), the number of training epochs (400), and the activation function (Gaussian Error Linear Unit). The dimension of the hidden layer was set to 256. The experiment was run 10 times for each set with different random seeds, and the average results are reported in Figure 2. We conclude that the $\alpha$ and $\beta$ hyperparameters do not have a large impact on the CFC model.

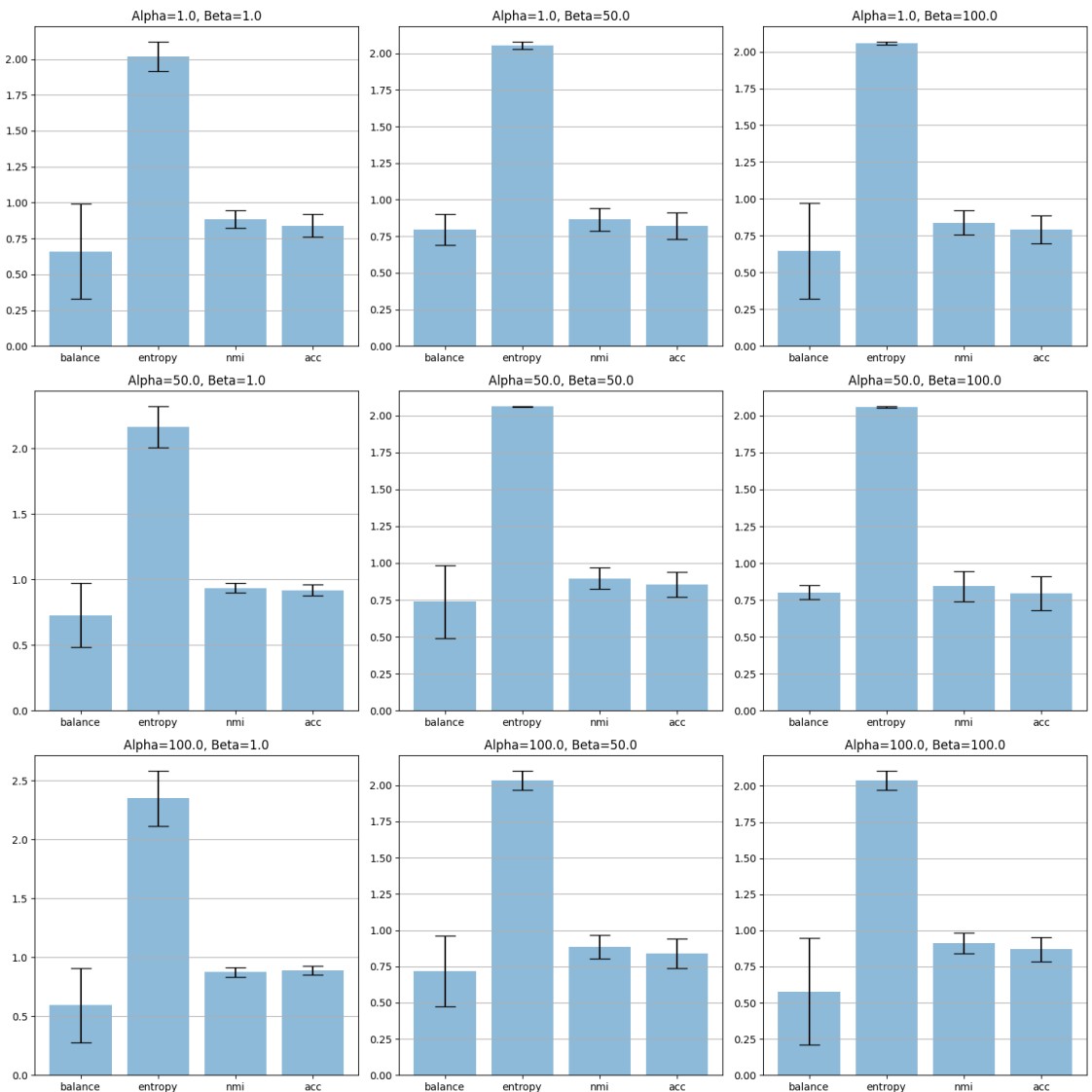

Figure 2: Results on on the effects of $\alpha$ and $\beta$ hyperparameters on the CFC model. To measure performance, the Balance, Entropy, Normalized Mutual Information (NMI) and Unsupervised Accuracy (ACC) metrics were used.

| Algorithm | Metric | MTFL | | | uncropped Yale | | |
|---|---|---|---|---|---|---|---|
| | | Pre-Attack | Post-Attack | Change (%) | Pre-Attack | Post-Attack | Change (%) |
| CFC | Balance | 0.583 | 0.571 | (-)2.058 | 0.191 | 0.440 | (+)130.4 |
| | Entropy | 0.639 | 0.635 | (-)0.626 | 7.921 | 7.520 | (-)5.062 |
| | NMI | 0.007 | 0.007 | (-)0.000 | 0.229 | 0.199 | (-)13.10 |
| | ACC | 0.600 | 0.606 | (+)1.000 | 0.136 | 0.117 | (-)13.97 |
| SFD | Balance | $0.971 \pm 0.000$ | $0.967 \pm 0.000$ | (-)0.412 | $0.115 \pm 0.116$ | $0.031 \pm 0.039$ | (-)73.04 |
| | Entropy | $0.692 \pm 0.000$ | $0.692 \pm 0.000$ | (-)0.000 | $12.00 \pm 0.194$ | $11.68 \pm 0.195$ | (-)2.765 |
| | NMI | $0.001 \pm 0.000$ | $0.000 \pm 0.000$ | (-)100.0 | $0.693 \pm 0.002$ | $0.687 \pm 0.005$ | (-)0.866 |
| | ACC | $0.529 \pm 0.000$ | $0.512 \pm 0.000$ | (-)3.214 | $0.404 \pm 0.003$ | $0.412 \pm 0.008$ | (+)1.980 |
| FSC | Balance | $0.992 \pm 0.000$ | $0.986 \pm 0.000$ | (-)0.605 | $0.000 \pm 0.000$ | $0.000 \pm 0.000$ | (-)0.000 |
| | Entropy | $0.693 \pm 0.000$ | $0.693 \pm 0.000$ | (-)0.000 | $11.11 \pm 0.030$ | $11.07 \pm 0.016$ | (-)0.360 |
| | NMI | $0.000 \pm 0.000$ | $0.000 \pm 0.000$ | (-)0.000 | $0.880 \pm 0.000$ | $0.879 \pm 0.000$ | (-)0.114 |
| | ACC | $0.546 \pm 0.000$ | $0.544 \pm 0.000$ | (-)0.366 | $0.769 \pm 0.001$ | $0.769 \pm 0.001$ | (-)0.000 |
| KFC | Balance | $0.870 \pm 0.143$ | $0.778 \pm 0.108$ | (-)10.58 | $0.774 \pm 0.295$ | $0.728 \pm 0.379$ | (-)5.943 |
| | Entropy | $0.684 \pm 0.019$ | $0.678 \pm 0.018$ | (-)0.877 | $0.503 \pm 0.160$ | $0.441 \pm 0.147$ | (-)12.33 |
| | NMI | $0.000 \pm 0.001$ | $0.000 \pm 0.000$ | (-)0.000 | $0.002 \pm 0.002$ | $0.001 \pm 0.002$ | (-)50.00 |
| | ACC | $0.669 \pm 0.011$ | $0.670 \pm 0.011$ | (+)0.149 | $0.032 \pm 0.001$ | $0.032 \pm 0.001$ | (-)0.000 |

Table 5: Results for pre-attack, post-attack (*black-box*), change between pre- and post-attack, and relative changes compared to the original study, when 15% group membership labels are switched for fair clustering algorithms CFC, SFD, FSC, and KFC and datasets *MTFL* and uncropped *Yale*. Results show the impact on fairness utility (Balance and Entropy) and clustering utility (NMI and ACC). Relative changes provide insights into how our changes between pre-attack and post-attack differ from those of the paper. The pre- and post-attack for CFC were run with one seed, leading to no standard deviation.

## 5 Discussion

Our study aimed to replicate key aspects of 'Robust Fair Clustering' (Chhabra et al., 2023), examining four algorithms across four datasets (see Section 3.5). We successfully confirmed Claims 2 and 3 of the original study, but only achieved partial replication for Claim 1, highlighting some variations in our experimental findings.

The results presented for Claim 1 (Section 4.1.1) partially confirm the effectiveness of the *black-box* attack in degrading the fairness performance of the clustering models. Across the four datasets and three fairness algorithms outlined in the baseline study, 75% of our experiments supported this claim, showcasing a significant reduction in fairness, with the Balance metric showing declines exceeding 30%. While our results largely mirrored those of the original study, there were notable exceptions: a slight increase, instead of a considerable decrease, for the Balance and Entropy metrics on the *MNIST-USPS* dataset using Scalable Fairlet Decomposition (SFD). In the baseline study, the fairness models were also evaluated on *DIGITS* and *Yale* datasets. Our experiments on these datasets reveal that the fairness attack significantly impairs the performance of the SFD model for both datasets, as detailed in Appendix E, Table 9. When subjected to attacks, three of the four datasets referenced in the baseline study show a substantial performance drop in the SFD model. The lack of a similar decline in the *MNIST-USPS* dataset suggests that the SFD model effectively copes with the attack on this dataset. The discrepancy in the SFD model's performance on *MNIST-USPS* between the baseline research and our analysis may be attributed to potential dataset preprocessing not disclosed in the baseline study or to the original results being incidental. Even though the outcomes for the SFD model on *MNIST-USPS* do not match those of the baseline study, the results from the other three datasets referenced in the baseline support the original claim. Similarly, on the *Office-31* dataset, the results for Fair K-Center (KFC) remained unchanged before and after the attack. Besides the reasons discussed above, this can be attributed to the unsupervised and unstable nature of clustering algorithms, which can result in singleton clustering or utilizing fewer clusters than specified (Ohl et al., 2022). Despite the discussed discrepancies, considering that the results of most of the experiments are similar to the findings in the baseline research, Claim 1 is partially confirmed.

Our analysis of Claim 2 (Section 4.1.2) aligns with the hypothesis that clustering models are prone to adversarial influence. As shown in the original paper, such attacks lead to significant fairness decreases across all fair clustering algorithms and datasets. While our observations showed some deviations from this pattern and some exceptions exist as previously discussed, they generally aligned with the original study

in demonstrating a consistent decline in fairness performance across the studied algorithms and datasets post-attack. Specifically, the fairness performance, as assessed by the Balance metric, experienced declines exceeding 30% in most instances, and in many cases, even exceeded 70%

The third claim (Section 4.1.3) states that the Contrastive Fair Clustering (CFC) algorithm exhibits high resilience against the proposed fairness attack. Consistent with the findings reported in the paper, our analysis revealed that the CFC algorithm demonstrated superior performance in terms of fairness utility and clustering performance compared to other state-of-the-art fair clustering algorithms. An exception was observed where the SFD algorithm exhibited a marginal improvement in performance following the attack on the *MNIST-USPS* dataset, as previously mentioned. This dataset represents the sole instance among the four evaluated in the baseline study where such an anomaly was noted. While there may exist some variations in the specific values of post-attack fairness compared to pre-attack fairness, our analysis generally confirms the superior performance of the CFC algorithm in defending against fairness attacks when compared to other algorithms. Thus, our findings affirm the robustness and effectiveness of the CFC algorithm in preserving fairness and performance under adversarial conditions.

Additionally, our study incorporated new evaluation metrics as outlined in Section 4.2.1, enhancing our understanding of model performance and guiding the development of new attack methods. Particularly noteworthy was the Minimum Cluster Ratio, which exhibited substantial shifts in several experiments, contrasting with other additional metrics that remained relatively stable pre- and post-attack. This suggests that they may not be suitable targets for fairness attacks, though further investigation is needed to confirm this hypothesis. Equally notable were the Silhouette Difference and Silhouette Score metrics. Initially introduced to assess fairness and clustering quality, respectively, these metrics played a crucial role in revealing the KFC algorithm's tendency towards singleton clustering, thereby enhancing the transparency of the model evaluations.

Our study was further extended by including two new datasets, as detailed in Section 4.2.1, to assess the adaptability of models across different data distributions. In the uncropped *Yale* dataset, CFC consistently outperformed its counterparts. For *MTFL*, the attacks were not very effective even for the KFC, FSC, and SFD algorithms. This could be attributed to the dataset being simplified to just two categories post-processing, making it relatively straightforward for the clustering models to effectively group the data into two clusters, even with 15% of the labels switched. Further investigation is necessary to fully understand the underlying reasons for this outcome.

Moreover, we conducted additional investigations to assess the generalizability of the model under diverse attack scenarios. The investigations included attacking different evaluation metrics other than the ones used in the baseline research, as well as a combined attack. The findings indicated that the performance of CFC remained consistently stable, even under these new forms of attack. This suggests that the current model demonstrates robustness against attacks designed in a similar way as the attack to the evaluation metric Balance in baseline research. Future studies involving more innovative attacks might offer opportunities to enhance and broaden the model's defensive capabilities.

Finally, the findings from our ablation study, particularly the minimal influence of the $\alpha$ and $\beta$ hyperparameters on the CFC model, underscore the robustness of the model's design. This resilience against hyperparameter fluctuations highlights the model's adaptability and efficiency in maintaining fairness and clustering quality across varied settings.

**Limitations and Future Work:** The primary constraints of our study were limited computational resources and time, leading us to use a restricted set of seeds in defense experiments and a narrow scope in our grid search. Further work could include expanding the grid search and running the defense experiments with a full set of seeds. Expanding the grid search parameters could potentially unveil a more effective attack strategy, aligning with our ultimate objective of identifying potent attacks. A particularly valuable addition to this study would be the development of a novel attack approach that can significantly challenge the robustness of the CFC algorithm, thereby pushing the boundaries of our current understanding of defense mechanisms in fair clustering.

**Environmental Impact:** Experiments were conducted using a computer cluster located in Amsterdam, which has a carbon efficiency of 0.4590 kgCO$_2$eq/kWh. A cumulative of 130 hours of computation was performed on hardware of type A100, and 80 hours on AMD Ryzen 7 4800H CPU. Total emissions are estimated to be 19.73 kgCO$_2$eq. Estimations were conducted using the Machine Learning Impact calculator presented in (Lacoste et al., 2019).

**Broader Impact:** Recent studies have shown that algorithmic decision-making may be inherently prone to unfairness, even when there is no intention for it (Pessach & Shmueli, 2022). This study contributes to the importance of fairness in clustering algorithms and their resilience against adversarial attacks. Our findings not only highlight the vulnerabilities of ML models to adversarial attacks as shown in existing literature (Ma et al., 2019), but also contribute to the verification of more robust defense mechanisms against such fairness attacks. Furthermore, by incorporating additional evaluation metrics and datasets, our research offers a more in-depth exploration of the fair clustering model's adaptability and resilience which is beneficial to further research. Through this research, we aim to highlight our study's role in advancing the technical robustness of AI technologies, contributing to a more equitable and sustainable future in the field of ML.

### 5.1 What was easy

The original paper provided the necessary information on the majority of hyperparameter values required to reproduce the experiments, and the publically available repository was well-documented with insightful comments. This made it straightforward to refactor the code and understand the idea of the proposed method. While it required some effort to comprehend and adapt the implementation structure, the overall time invested in executing the experiments successfully was reasonable.

### 5.2 What was difficult

One minor inconvenience was that certain dependencies needed slight adjustments to align better with current popular environments. Moreover, incorporating the KFC algorithm required the installation of IBM-CPLEX, used as an external solver via PuLP. Another potential concern was the lack of clarity in the baseline research regarding the acquisition of pre-computed labels and index files, interfering with the expansion to additional datasets and evaluation metrics for assessing the robustness of the conclusions.

### 5.3 Communication with original authors

There was no direct communication with the original authors throughout the replication effort. Later advice on hyperparameters for the KFC algorithm from the authors did not impact the noted singleton clustering behavior.

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

## A  Hyperparameters

- **SFD algorithm:** The values of $p$ and $q$ parameters are set to efficiently balance the trade-off between clustering performance and fairness utility. Specifically, we use $p = 2$ and $q = 5$ for most datasets, except for the *Inverted UCI DIGITS* dataset where $p = 1$ and $q = 5$.

- **FSC algorithm:** we employ the nearest neighbors approach to construct the input graph, setting the number of neighbors to 3 for all datasets.

- **KFC algorithm:** we utilize a parameter $\delta = 0.1$ as specified in the original implementation.

- **CFC algorithm:** consistency in hyperparameters is maintained across all datasets, with a fixed number of basic partitions ($r$) set at 100, a temperature parameter ($\tau$) of 2 for the contrastive loss ($L_c$), and a dropout of 0.6 in hidden layers.

Furthermore, the activation function employed is the Gaussian Error Linear Unit (GELU) (Hendrycks & Gimpel, 2016), while the fair clustering algorithm utilized for generating $J$ for structural preservation loss ($L_p$) is SFD with default parameters, chosen for its faster runtime compared to other fair clustering algorithms. The dimension of the hidden layer is set to 256 for all datasets except *Inverted UCI DIGITS*, which has only 64 features, thus necessitating a hidden layer dimension of 36.

Optimization of other hyperparameters for different datasets to enhance fairness performance is conducted using a grid-based search technique and are shown in Table 6.

| Dataset | $R$ | $\alpha$ | $\beta$ |
|---|---|---|---|
| *MNIST-USPS* | 2 | 100 | 25 |
| *Office-31* | 1 | 1 | 100 |
| *Inverted UCI DIGITS* | 2 | 10 | 50 |
| *Extended Yale face B* | 2 | 50 | 10 |

Table 6: Summary of hyperparameters used in our experimentation.

## B  KFC: Comparison of Balance and Entropy Attacks

The original paper utilized the Fair K-Center (KFC) algorithm, but the code for this algorithm was absent in the jupyter notebooks provided by the authors. Consequently, we had to make arbitrary decisions regarding each experiment's budget and attack strategy. In Table 7, we present the results for the Balance and Entropy attacks for KFC on all datasets. Notably, these results consistently fall within a very similar range and, in some instances, are even identical across datasets. Upon further investigation, it was discovered that KFC clustered all data points into the same cluster, likely explaining the uniformity of the results.

| Dataset | Attack | Metric | | | |
|---|---|---|---|---|---|
| | | Balance | Entropy | NMI | ACC |
| *MNIST-USPS* | Balance | $0.350 \pm 0.299$ | $1.242 \pm 0.418$ | $0.027 \pm 0.017$ | $0.144 \pm 0.010$ |
| | Entropy | $0.350 \pm 0.299$ | $1.202 \pm 0.350$ | $0.028 \pm 0.019$ | $0.145 \pm 0.012$ |
| *Office-31* | Balance | $0.971 \pm 0.067$ | $0.401 \pm 0.135$ | $0.001 \pm 0.003$ | $0.050 \pm 0.000$ |
| | Entropy | $0.971 \pm 0.067$ | $0.401 \pm 0.135$ | $0.001 \pm 0.003$ | $0.050 \pm 0.000$ |
| *DIGITS* | Balance | $0.313 \pm 0.203$ | $3.154 \pm 0.244$ | $0.056 \pm 0.007$ | $0.174 \pm 0.017$ |
| | Entropy | $0.375 \pm 0.209$ | $3.133 \pm 0.220$ | $0.028 \pm 0.019$ | $0.175 \pm 0.016$ |
| *Yale* | Balance | $0.834 \pm 0.322$ | $0.668 \pm 0.578$ | $0.003 \pm 0.007$ | $0.030 \pm 0.001$ |
| | Entropy | $0.845 \pm 0.326$ | $0.845 \pm 0.326$ | $0.602 \pm 0.517$ | $0.030 \pm 0.001$ |

Table 7: Comparison of Balance and Entropy attacks for the KFC algorithm for *MNIST-USPS*, *Office-31*, *Inverted UCI DIGITS* (*DIGITS*), and *Extended Yale face B* (*Yale*) datasets. Results show the impact on fairness utility (Balance and Entropy) and clustering utility (NMI and ACC).

## C  Definitions for Metrics

**Normalized Mutual Information** (NMI) is a normalized version of the mutual information metric. It can be defined as Equation 1, with the mutual information metric (Shannon, 1948) represented by $I$, Shannon's entropy represented by $E$, the cluster assignment labels represented by $L$, and the ground truth labels represented by $Y$:

$$\text{NMI} = \frac{I(Y, L)}{\frac{1}{2}\left[E(Y) + E(L)\right]} \tag{1}$$

**Accuracy** (ACC) is equivalent to the classic accuracy for classification. A mapping function $\rho$ is utilized to compute all feasible mappings between true labels and predicted cluster labels for some $m$ samples. It defines $Y_i$ as the true labels and $L_i$ as the predicted cluster labels and forms the following Equation:

$$\text{ACC} = \max_{\rho} \frac{\sum_{i=1}^{m} 1\{Y_i = \rho(L_i)\}}{m} \tag{2}$$

**Balance** lies between 0 (least fair) and 1 (most fair). Suppose that the model contains $m$ protected groups for a given dataset $X$. Equation 3 denotes $r_X^g$ as the proportion of samples of the dataset belonging to protected group g and $r_k^g$ as the proportion of samples in cluster k $\in$ [K] belonging to protected group $g$:

$$\text{Balance} = \min_{k \in [K], g \in [m]} \min\left\{\frac{r_X^g}{r_k^g}, \frac{r_k^g}{r_X^g}\right\} \tag{3}$$

**Entropy** is similar to Balance, where higher values of Entropy indicate that clusters have more fairness. In Equation 4, $N_{k,g}$ is used to represent the set containing the samples of the dataset $X$ that belong to both the cluster $k \in [K]$ and the protected group $g$. Besides, $n_k$ is used to denote the number of samples in cluster $k$:

$$\text{Entropy}(g) = -\sum_{k \in [K]} \frac{|N_{k,g}|}{n_k} \log \frac{|N_{k,g}|}{n_k} \tag{4}$$

# D  Attack and Defense Results on MNIST-USPS and Office-31 with Relative Changes

| Algorithm | Metrics | MNIST-USPS | | | | | | |
| --- | --- | --- | --- | --- | --- | --- | --- | --- |
| | | Pre-Attack | Post-Attack | Change (%) | Relative Change (%) | Random Attack | Change (%) | Relative Change (%) |
| SFD | Balance | 0.282 ± 0.001 | 0.300 ± 0.001 | **(+)6.382** | (+)106.4 | 0.330 ± 0.001 | (+)17.02 | (+)117.0 |
| | Entropy | 3.063 ± 0.151 | 3.104 ± 0.001 | **(+)1.339** | (+)1.028 | 3.147 ± 0.000 | (+)2.742 | (+)106.3 |
| | NMI | 0.315 ± 0.000 | 0.358 ± 0.000 | (+)13.65 | (+)348.7 | 0.346 ± 0.000 | (+)9.841 | (+)268.3 |
| | ACC | 0.419 ± 0.000 | 0.473 ± 0.000 | (+)12.89 | (+)211.7 | 0.456 ± 0.000 | (+)8.831 | (+)171.5 |
| FSC | Balance | 0.000 ± 0.000 | 0.000 ± 0.000 | (-)100.0 | (-)0.000 | 0.000 ± 0.000 | (-)100.0 | (-)0.000 |
| | Entropy | 0.327 ± 0.000 | 0.241 ± 0.001 | (-)26.30 | (-)206.7 | 0.301 ± 0.001 | (-)7.951 | (-)204.6 |
| | NMI | 0.549 ± 0.000 | 0.543 ± 0.000 | (-)1.093 | (-)35.44 | 0.538 ± 0.000 | (-)2.004 | (-)508.1 |
| | ACC | 0.450 ± 0.000 | 0.454 ± 0.000 | (+)0.889 | (-)54.90 | 0.443 ± 0.000 | (-)1.556 | (-)203.0 |
| KFC | Balance | 0.557 ± 0.324 | 0.350 ± 0.299 | (-)37.16 | (-)39.86 | 0.724 ± 0.117 | (+)30.20 | (+)280.6 |
| | Entropy | 1.355 ± 0.374 | 1.202 ± 0.351 | (-)11.29 | (-)11.56 | 1.417 ± 0.417 | (+)4.576 | (+)1,317 |
| | NMI | 0.000 ± 0.000 | 0.000 ± 0.000 | (-)100.0 | (-)1,820 | 0.000 ± 0.000 | (-)100.0 | (-)1,976 |
| | ACC | 0.147 ± 0.000 | 0.146 ± 0.000 | (-)0.680 | (-)114.8 | 0.145 ± 0.000 | (-)1.361 | (-)139.8 |
| Algorithm | Metrics | Office-31 | | | | | | |
| | | Pre-Attack | Post-Attack | Change (%) | Relative Change (%) | Random Attack | Change (%) | Relative Change (%) |
| SFD | Balance | 0.546 ± 0.000 | 0.158 ± 0.000 | (-)71.06 | (+)18.52 | 0.359 ± 0.120 | (-)34.25 | (+)39.21 |
| | Entropy | 10.00 ± 0.000 | 9.783 ± 0.001 | (-)2.170 | (+)34.42 | 9.903 ± 0.001 | (-)0.970 | (+)62.42 |
| | NMI | 0.888 ± 0.000 | 0.861 ± 0.000 | (-)3.041 | (+)26.01 | 0.860 ± 0.000 | (-)3.153 | (-)334.3 |
| | ACC | 0.841 ± 0.000 | 0.765 ± 0.000 | (-)9.037 | (+)3.831 | 0.769 ± 0.000 | (-)8.561 | (-)194.4 |
| FSC | Balance | 0.000 ± 0.000 | 0.000 ± 0.000 | (-)100.0 | (-)0.000 | 0.211 ± 0.211 | **(+)100.0** | (-)1.903 |
| | Entropy | 9.164 ± 0.119 | 9.383 ± 0.301 | **(+)2.390** | (+)339.7 | 9.628 ± 0.213 | (+)5.063 | (+)2,346 |
| | NMI | 0.652 ± 0.000 | 0.682 ± 0.000 | (+)4.601 | (+)25.74 | 0.685 ± 0.000 | (+)5.061 | (+)36.38 |
| | ACC | 0.390 ± 0.000 | 0.438 ± 0.000 | (+)12.31 | (+)24.30 | 0.436 ± 0.000 | (+)18.72 | (+)112.3 |
| KFC | Balance | 0.971 ± 0.001 | 0.971 ± 0.001 | **(-)0.000** | (+)100.0 | 0.971 ± 0.001 | (-)0.000 | (+)100.0 |
| | Entropy | 0.401 ± 0.135 | 0.401 ± 0.135 | **(-)0.000** | (+)100.0 | 0.401 ± 0.135 | (-)0.000 | (+)100.0 |
| | NMI | 0.000 ± 0.000 | 0.000 ± 0.000 | (-)100.0 | (-)206.0 | 0.000 ± 0.000 | (-)100.0 | (-)62,400 |
| | ACC | 0.001 ± 0.000 | 0.001 ± 0.000 | (-)0.000 | (-)100.0 | 0.001 ± 0.000 | (-)0.000 | (+)100.0 |

Table 8: Results for pre-attack, post-attack (*black-box*), random attack, change between pre- and post-attack / random attack, and relative changes compared to the original study, when 15% group membership labels are switched for fair clustering algorithms SFD, FSC, and KFC and datasets *MNIST-USPS* and *Office-31*. Results show the impact on fairness utility (Balance and Entropy) and clustering utility (NMI and ACC). Relative changes provide insights into how our changes between pre-attack and post-attack / random attack differ from those of the paper.

# E  Attack Results on DIGITS and Yale Datasets

| Algorithms | Metrics | DIGITS | | | | | | |
| --- | --- | --- | --- | --- | --- | --- | --- | --- |
| | | Pre-Attack | Post-Attack | Change (%) | Relative Change (%) | Random Attack | Change (%) | Relative Change (%) |
| SFD | Balance | 0.000 ± 0.000 | 0.000 ± 0.000 | (-)100.0 | (-)0.000 | 0.000 ± 0.000 | (-)100.0 | (-)0.000 |
| | Entropy | 2.921 ± 0.000 | 0.000 ± 0.000 | (-)100.0 | (-)0.000 | 0.000 ± 0.000 | (-)100.0 | (-)0.000 |
| | NMI | 0.278 ± 0.000 | 0.393 ± 0.000 | (+)41.37 | (+)28.44 | 0.393 ± 0.000 | (+)41.37 | (+)21.04 |
| | ACC | 0.399 ± 0.000 | 0.436 ± 0.000 | (+)9.273 | (+)70.46 | 0.436 ± 0.000 | (+)9.273 | (+)27.38 |
| FSC | Balance | 0.000 ± 0.000 | 0.000 ± 0.000 | (-)100.0 | (-)0.000 | 0.000 ± 0.000 | (-)100.0 | (-)0.000 |
| | Entropy | 0.346 ± 0.000 | 0.342 ± 0.000 | (-)1.156 | NA | 0.346 ± 0.000 | (-)0.000 | (-)100.0 |
| | NMI | 0.564 ± 0.000 | 0.561 ± 0.000 | (-)0.532 | NA | 0.564 ± 0.000 | (-)0.000 | (+)100.0 |
| | ACC | 0.313 ± 0.000 | 0.316 ± 0.000 | (+)0.958 | NA | 0.314 ± 0.000 | (+)0.319 | (-)9.117 |
| KFC | Balance | 0.707 ± 0.132 | 0.375 ± 0.209 | (-)46.96 | (+)33.99 | 0.394 ± 0.306 | (-)44.27 | (-)1.840 |
| | Entropy | 3.376 ± 0.107 | 3.133 ± 0.220 | (-)7.198 | (-)5.620 | 3.210 ± 0.197 | (-)4.917 | (-)16.63 |
| | NMI | 0.001 ± 0.000 | 0.001 ± 0.000 | (-)0.000 | (+)100.0 | 0.001 ± 0.000 | (-)0.000 | (+)100.0 |
| | ACC | 0.174 ± 0.000 | 0.175 ± 0.000 | (+)0.575 | (+)143.1 | 0.174 ± 0.000 | (-)0.000 | (+)100.0 |
| Algorithms | Metrics | Yale | | | | | | |
| | | Pre-Attack | Post-Attack | Change (%) | Relative Change (%) | Random Attack | Change (%) | Relative Change (%) |
| SFD | Balance | 0.000 ± 0.000 | 0.000 ± 0.000 | (-)100.0 | (-)0.000 | 0.000 ± 0.000 | (-)100.0 | (-)0.000 |
| | Entropy | 3.969 ± 0.232 | 3.741 ± 0.193 | (-)5.745 | (+)46.85 | 4.326 ± 0.374 | (+)8.994 | (+)2,192 |
| | NMI | 0.160 ± 0.000 | 0.160 ± 0.000 | (-)0.000 | (-)100.0 | 0.164 ± 0.000 | (+)2.500 | (-)64.55 |
| | ACC | 0.001 ± 0.000 | 0.001 ± 0.000 | (-)0.000 | (-)100.0 | 0.001 ± 0.000 | (-)0.000 | (-)100.0 |
| FSC | Balance | 0.000 ± 0.000 | 0.000 ± 0.000 | (-)100.0 | (-)0.000 | 0.000 ± 0.000 | (-)100.0 | (-)0.000 |
| | Entropy | 3.402 ± 0.001 | 3.261 ± 0.001 | (-)4.145 | (-)42.34 | 3.488 ± 0.123 | (+)2.528 | (+)93.87 |
| | NMI | 0.367 ± 0.000 | 0.365 ± 0.000 | (-)0.545 | (-)2.830 | 0.366 ± 0.000 | (-)0.272 | (-)129.5 |
| | ACC | 0.273 ± 0.000 | 0.274 ± 0.000 | (+)0.366 | (+)314.0 | 0.273 ± 0.000 | (-)0.000 | (+)100.0 |
| KFC | Balance | 0.800 ± 0.400 | 0.800 ± 0.400 | (-)0.000 | (-)100.0 | 0.800 ± 0.400 | (-)0.000 | (-)100.0 |
| | Entropy | 0.344 ± 0.000 | 0.344 ± 0.000 | (-)0.000 | (-)100.0 | 0.344 ± 0.000 | (-)0.000 | (-)100.0 |
| | NMI | 0.000 ± 0.000 | 0.000 ± 0.000 | (-)0.000 | (-)864.5 | 0.000 ± 0.000 | (-)100.0 | (-)709.0 |
| | ACC | 0.241 ± 0.000 | 0.241 ± 0.000 | (-)0.000 | (+)100.0 | 0.241 ± 0.000 | (-)0.000 | (+)100.0 |

Table 9: Results for pre-attack, post-attack (*black-box*), random attack, change between pre- and post-attack / random attack ,and relative changes compared to the original study, when 15% group membership labels are switched for fair clustering algorithms SFD, FSC, and KFC and datasets *Inverted UCI DIGITS* (*DIGITS*) and *Extended Yale face B* (*Yale*). Results show the impact on fairness utility (Balance and Entropy) and clustering utility (NMI and ACC). Relative changes provide insights into how our changes between pre-attack and post-attack / random attack differ from those of the paper.

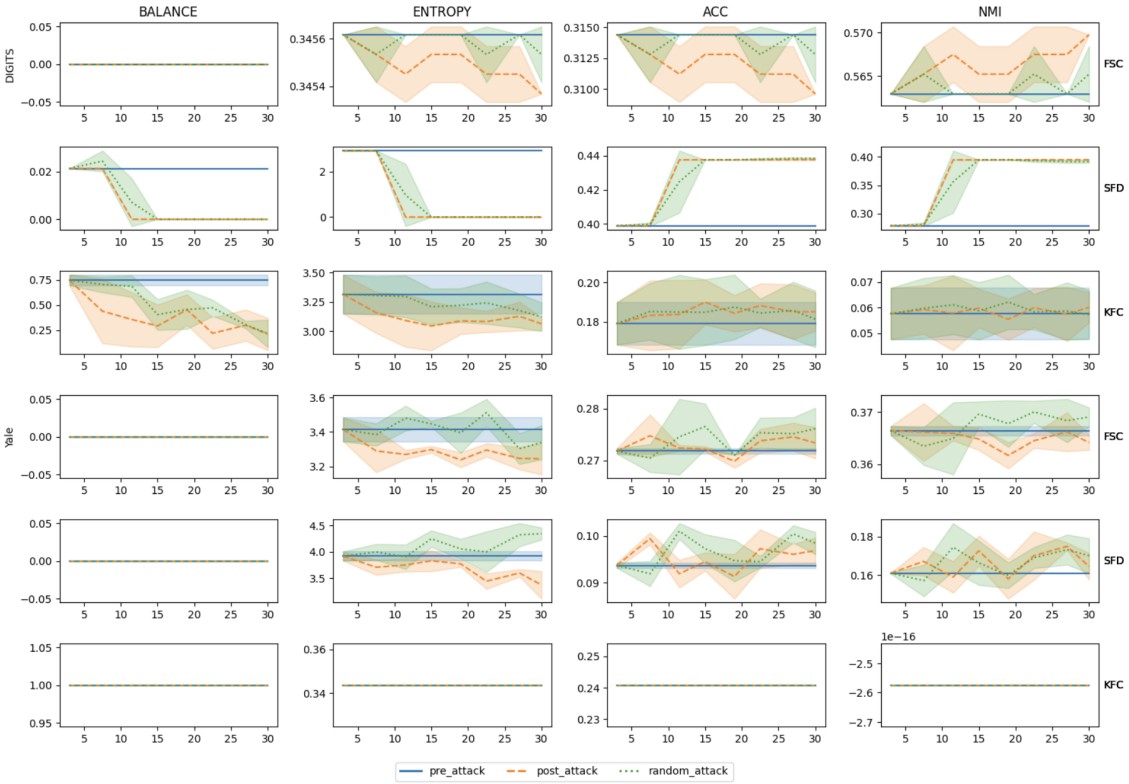

Figure 3: Pre-attack, post-attack (*black-box*) and random attack results on fairness utility (Balance and Entropy) and clustering utility (ACC and NMI) for *Inverted UCI DIGITS (DIGITS)* and *Extended Yale face B (Yale)* (x-axis: % of samples attacker can poison).

## F  Defense Results on DIGITS and Yale Datasets

| Algorithms | Metrics | DIGITS | | | | Yale | | | |
|---|---|---|---|---|---|---|---|---|---|
| | | Pre-Attack | Post-Attack | Change (%) | Relative Change (%) | Pre-Attack | Post-Attack | Change (%) | Relative Change (%) |
| CFC | Balance | 0.139±0.011 | 0.145±0.047 | (+)4.50 | (-)94.61 | 0.001±0.001 | 0.219±0.001 | (+)21,800 | (+)29,873 |
| | Entropy | 1.978±0.051 | 1.996±0.079 | (+)0.94 | (-)86.09) | 6.015±0.193 | 6.813±1.010 | (+)13.27 | (+)0.378 |
| | NMI | 0.226±0.000 | 0.290±0.000 | (+)28.32 | (-)269.4 | 0.150±0.000 | 0.144±0.000 | (-)4.000 | (-)119.8 |
| | ACC | 0.283±0.000 | 0.293±0.000 | (+)3.534 | (+)157.6 | 0.150±0.000 | 0.144±0.000 | (-)4.000 | (-)142.6 |
| SFD | Balance | 0.000±0.000 | 0.000±0.000 | (-)100.0 | (-)0.000 | 0.000±0.000 | 0.000±0.000 | (-)100.0 | (-)0.000 |
| | Entropy | 2.921±0.000 | 0.000±0.000 | (-)100.0 | (-)0.000 | 3.969±0.232 | 3.741±0.193 | (-)5.745 | (+)46.85 |
| | NMI | 0.278±0.000 | 0.393±0.000 | (+)41.37 | (+)28.44 | 0.160±0.000 | 0.160±0.000 | (-)0.000 | (-)100.0 |
| | ACC | 0.399±0.000 | 0.436±0.000 | (+)9.273 | (+)70.46 | 0.001±0.000 | 0.001±0.000 | (-)0.000 | (-)100.0 |
| FSC | Balance | 0.000±0.000 | 0.000±0.000 | (-)100.0 | (-)0.000 | 0.000±0.000 | 0.000±0.000 | (-)100.0 | (-)0.000 |
| | Entropy | 0.346±0.000 | 0.342±0.000 | (-)1.156 | N/A* | 3.402±0.001 | 3.261±0.001 | (-)4.145 | (-)42.34 |
| | NMI | 0.564±0.000 | 0.561±0.000 | (-)0.532 | N/A* | 0.367±0.000 | 0.365±0.000 | (-)0.545 | (-)2.830 |
| | ACC | 0.313±0.000 | 0.316±0.000 | (+)0.958 | N/A* | 0.273±0.000 | 0.274±0.000 | (+)0.366 | (+)314.0 |
| KFC | Balance | 0.707±0.132 | 0.375±0.209 | (-)46.99 | (+)33.99 | 0.800±0.400 | 0.800±0.400 | (-)0.000 | (-)100.0 |
| | Entropy | 3.376±0.107 | 3.133 ± 0.22 | (-)7.198 | (-)5.620 | 0.344±0.000 | 0.344±0.000 | (-)0.000 | (-)100.0 |
| | NMI | 0.001±0.000 | 0.001±0.000 | (-)0.000 | (+)100.0 | 0.000±0.000 | 0.000±0.000 | (-)100.0 | (-)864.5 |
| | ACC | 0.174±0.000 | 0.175±0.000 | (+)0.575 | (+)143.1 | 0.241±0.000 | 0.241±0.000 | (-)0.000 | (+)100.0 |

Table 10: Results for pre-attack, post-attack (*black-box*), random attack, change between pre- and post-attack, and relative changes compared to the original study, when 15% group membership labels are switched for fair clustering algorithms SFD, FSC, and KFC and datasets *Inverted UCI DIGITS (DIGITS)* and *Extended Yale face B (Yale)*. Results show the impact on fairness utility (Balance and Entropy) and clustering utility (NMI and ACC). Relative changes provide insights into how our changes between pre-attack and post-attack / random attack differ from those of the paper. The N/A values in the relative changes column indicate instances where the change in the original paper was 0%, making division by 0 impossible.

# G   Analyzing Overall Adversial Robustness of CFC

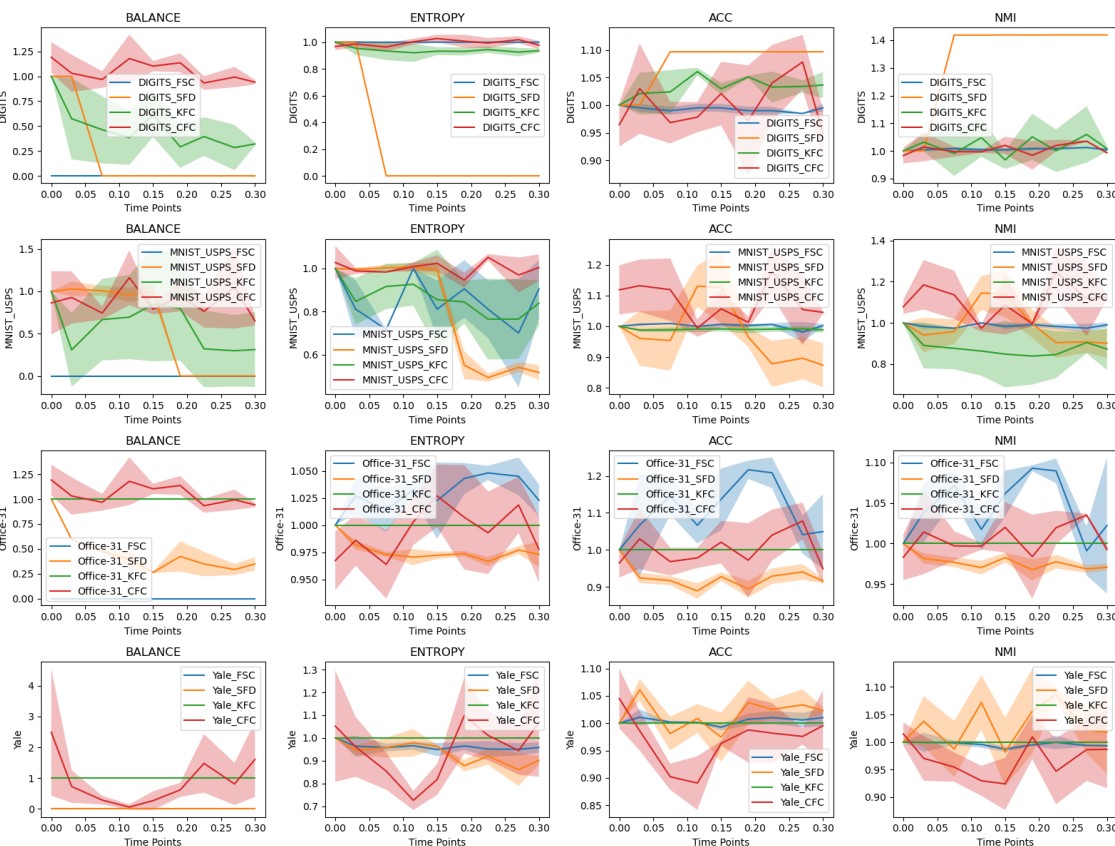

Figure 4: Pre-attack and post-attack (*black-box*) ratio trends for FSC, SFD, KFC, and CFC on fairness utility (Balance and Entropy) and clustering utility (ACC and NMI) for *MNIST-USPS* and *Office-31* (we do not plot curves for which pre-attack values are 0) (x-axis: % of samples attacker can poison).

## H Extra Metrics for DIGITS and Yale Datasets

| Algorithms | Metrics | DIGITS | | | Yale | | |
|---|---|---|---|---|---|---|---|
| | | Pre-Attack | Post-Attack | Random Attack | Pre-Attack | Post-Attack | Random Attack |
| SFD | Min. Cluster Ratio | $0.395 \pm 0.001$ | $0.000 \pm 0.000$ | $0.000 \pm 0.000$ | $0.000 \pm 0.000$ | $0.000 \pm 0.000$ | $0.000 \pm 0.000$ |
| | Cluster L1 | $0.344 \pm 0.001$ | $1.000 \pm 0.000$ | $1.000 \pm 0.000$ | $0.797 \pm 0.009$ | $0.804 \pm 0.014$ | $0.779 \pm 0.015$ |
| | Cluster KL | $0.722 \pm 0.002$ | $\infty^* \pm nan^*$ | $\infty^* \pm nan^*$ | $\infty^* \pm nan^*$ | $\infty^* \pm nan^*$ | $\infty^* \pm nan^*$ |
| | Silhouette diff | $-0.060 \pm 0.000$ | $-0.090 \pm 0.124$ | $-0.060 \pm 0.142$ | $-0.002 \pm 0.005$ | $-0.008 \pm 0.004$ | $-0.008 \pm 0.005$ |
| | Entropy Group A | $2.271 \pm 0.000$ | $1.739 \pm 0.869$ | $1.525 \pm 0.998$ | $1.788 \pm 0.100$ | $1.816 \pm 0.180$ | $1.752 \pm 0.166$ |
| | Entropy Group B | $1.983 \pm 0.002$ | $0.435 \pm 0.869$ | $0.652 \pm 0.996$ | $3.505 \pm 0.016$ | $3.491 \pm 0.016$ | $3.500 \pm 0.014$ |
| | ARI | $0.157 \pm 0.000$ | $0.094 \pm 0.001$ | $0.094 \pm 0.000$ | $0.007 \pm 0.001$ | $0.009 \pm 0.006$ | $0.008 \pm 0.006$ |
| | Silhouette score | $-0.072 \pm 0.000$ | $0.322 \pm 0.003$ | $0.321 \pm 0.001$ | $0.065 \pm 0.004$ | $0.060 \pm 0.004$ | $0.061 \pm 0.006$ |
| FSC | Min. Cluster Ratio | $0.000 \pm 0.000$ | $0.000 \pm 0.000$ | $0.000 \pm 0.000$ | $0.000 \pm 0.000$ | $0.000 \pm 0.000$ | $0.000 \pm 0.000$ |
| | Cluster L1 | $0.404 \pm 0.000$ | $0.404 \pm 0.000$ | $0.419 \pm 0.044$ | $0.760 \pm 0.025$ | $0.744 \pm 0.052$ | $0.762 \pm 0.013$ |
| | Cluster KL | $\infty^* \pm nan^*$ | $\infty^* \pm nan^*$ | $\infty^* \pm nan^*$ | $\infty^* \pm nan^*$ | $\infty^* \pm nan^*$ | $\infty^* \pm nan^*$ |
| | Silhouette diff | $-0.017 \pm 0.000$ | $-0.017 \pm 0.000$ | $-0.018 \pm 0.005$ | $0.004 \pm 0.004$ | $0.008 \pm 0.005$ | $0.002 \pm 0.005$ |
| | Entropy Group A | $1.028 \pm 0.006$ | $1.024 \pm 0.009$ | $1.033 \pm 0.017$ | $2.966 \pm 0.021$ | $2.918 \pm 0.087$ | $2.984 \pm 0.038$ |
| | Entropy Group B | $1.281 \pm 0.000$ | $1.281 \pm 0.000$ | $1.322 \pm 0.124$ | $2.280 \pm 0.024$ | $2.257 \pm 0.032$ | $2.272 \pm 0.033$ |
| | ARI | $0.156 \pm 0.000$ | $0.156 \pm 0.000$ | $0.158 \pm 0.008$ | $0.062 \pm 0.002$ | $0.058 \pm 0.007$ | $0.062 \pm 0.002$ |
| | Silhouette score | $-0.139 \pm 0.000$ | $-0.140 \pm 0.000$ | $-0.138 \pm 0.003$ | $-0.009 \pm 0.012$ | $-0.020 \pm 0.017$ | $-0.006 \pm 0.005$ |
| KFC | Min. Cluster Ratio | $0.683 \pm 0.128$ | $0.426 \pm 0.197$ | $0.494 \pm 0.246$ | $0.640 \pm 0.320$ | $0.665 \pm 0.276$ | $0.665 \pm 0.276$ |
| | Cluster L1 | $0.094 \pm 0.011$ | $0.120 \pm 0.011$ | $0.117 \pm 0.011$ | $0.001 \pm 0.001$ | $0.001 \pm 0.002$ | $0.001 \pm 0.002$ |
| | Cluster KL | $0.022 \pm 0.005$ | $\infty^* \pm nan^*$ | $\infty^* \pm nan^*$ | $\infty^* \pm nan^*$ | $\infty^* \pm nan^*$ | $\infty^* \pm nan^*$ |
| | Silhouette diff | $-0.014 \pm 0.021$ | $-0.034 \pm 0.052$ | $-0.026 \pm 0.051$ | N/A$^*$ | N/A$^*$ | N/A$^*$ |
| | Entropy Group A | $1.82 \pm 0.169$ | $1.869 \pm 0.157$ | $1.850 \pm 0.185$ | $0.015 \pm 0.029$ | $0.015 \pm 0.029$ | $0.014 \pm 0.030$ |
| | Entropy Group B | $1.82 \pm 0.168$ | $1.853 \pm 0.173$ | $1.831 \pm 0.194$ | $0.014 \pm 0.029$ | $0.016 \pm 0.031$ | $0.014 \pm 0.028$ |
| | ARI | $0.022 \pm 0.009$ | $0.023 \pm 0.008$ | $0.023 \pm 0.008$ | $0.000 \pm 0.000$ | $0.000 \pm 0.000$ | $0.000 \pm 0.000$ |
| | Silhouette score | $-0.07 \pm 0.032$ | $-0.152 \pm 0.059$ | $-0.130 \pm 0.048$ | N/A$^*$ | N/A$^*$ | N/A$^*$ |

Table 11: Results for pre-attack, post-attack (*black-box*) and random attack, when 15% group membership labels are switched for fair clustering algorithms SFD, FSC, and KFC and datasets *Inverted UCI DIGITS* (*DIGITS*) and *Extended Yale face B* (*Yale*). Results show the impact on additional metrics, where N/A corresponds to uniform clustering, $\infty$ to infinite values, and *nan* to undefined values.

## I Additional Attack Methods Results

| Metric | Attack Balance | Attack Min. Cluster Ratio | Combined Attack |
|---|---|---|---|
| Balance | $0.149 \pm 0.004$ | $0.149 \pm 0.004$ | $\mathbf{0.144 \pm 0.011}$ |
| Entropy | $9.764 \pm 0.037$ | $9.764 \pm 0.037$ | $\mathbf{9.715 \pm 0.089}$ |
| NMI | $0.857 \pm 0.009$ | $0.857 \pm 0.009$ | $\mathbf{0.857 \pm 0.002}$ |
| ACC | $0.757 \pm 0.026$ | $0.757 \pm 0.026$ | $\mathbf{0.753 \pm 0.016}$ |
| Min. Cluster Ratio | $0.061, \pm 0.002$ | $0.061, \pm 0.002$ | $\mathbf{0.059 \pm 0.005}$ |
| Cluster L1 | $0.178 \pm 0.007$ | $0.178 \pm 0.007$ | $\mathbf{0.183 \pm 0.002}$ |
| Cluster KL | $0.099 \pm 0.009$ | $0.099 \pm 0.009$ | $\mathbf{0.104 \pm 0.006}$ |
| Silhouette diff | $-0.009 \pm 0.001$ | $-0.008 \pm 0.002$ | $-0.005 \pm 0.002$ |
| Entropy Group A | $3.291 \pm 0.016$ | $3.291 \pm 0.016$ | $\mathbf{3.287 \pm 0.035}$ |
| Entropy Group B | $\mathbf{3.357 \pm 0.010}$ | $\mathbf{3.357 \pm 0.010}$ | $3.360 \pm 0.009$ |
| ARI | $\mathbf{0.677 \pm 0.021}$ | $\mathbf{0.677 \pm 0.021}$ | $0.681 \pm 0.010$ |
| Silhouette Score | $\mathbf{0.153 \pm 0.006}$ | $\mathbf{0.153 \pm 0.006}$ | $0.157 \pm 0.003$ |

Table 12: Results of Additional Attack Methods: This table compares the performance of the original balance attack against the newly introduced Minimum Cluster Ratio attack and the Combined (Balance & Entropy) attack. For a consistent comparison, the results presented are based on the same three seeds that were utilized during the grid search. The most effective attack strategy for each metric, as indicated by the lowest value, is emphasized in bold.

## J   Defense Results on Additional Attack Methods

| Attack Type | Metric | MNIST-USPS | | Office-31 | |
|---|---|---|---|---|---|
| | | Pre-Attack | Post-Attack | Pre-Attack | Post-Attack |
| Minimum Cluster Ratio | Min. Cluster Ratio | 0.402 | 0.306 | 0.319 | 0.371 |
| | Cluster L1 | 0.271 | 0.238 | 0.180 | 0.139 |
| | Cluster KL | 0.270 | 0.192 | 0.093 | 0.079 |
| | Silhouette diff | $-0.030$ | $-0.022$ | $-0.008$ | $-0.008$ |
| | Entropy Group A | 2.052 | 2.022 | 2.787 | 2.760 |
| | Entropy Group B | 1.849 | 1.899 | 2.775 | 2.768 |
| | ARI | 0.138 | 0.193 | 0.438 | 0.415 |
| | Silhouette score | 0.001 | 0.015 | 0.077 | 0.072 |
| Combined | Min. Cluster Ratio | 0.403 | 0.305 | 0.365 | 0.324 |
| | Cluster L1 | 0.214 | 0.256 | 0.155 | 0.180 |
| | Cluster KL | $\infty^*$ | $\infty^*$ | $\infty^*$ | 0.092 |
| | Silhouette diff | $-0.029$ | $-0.015$ | $-0.012$ | $-0.017$ |
| | Entropy Group A | 2.019 | 2.127 | 2.845 | 2.812 |
| | Entropy Group B | 1.877 | 1.937 | 2.810 | 2.870 |
| | ARI | 0.159 | 0.204 | 0.399 | 0.444 |
| | Silhouette score | 0.002 | 0.014 | 0.053 | 0.091 |

Table 13: Results for pre-attack and post-attack when 15% group membership labels are switched for defense algorithm CFC and datasets *MNIST-USPS* and *Office-31*. Results show the impact on additional metrics, where $\infty$ corresponds to infinite values. Experiments are run once because of the consumed GPU hours and the small standard deviations in other related experiments.

