# OpenReview forum: "Reproducibility Study of "Robust Fair Clustering: A Novel Fairness Attack and Defense Framework""
_TMLR — Accepted by TMLR_

### Review · Reviewer_XiUa · 2024-03-06

**Summary Of Contributions:**

This paper is a reproducibility study of "robust fair clustering," where the authors summarized five contribution points including one reproducibility and four extended works. I agree with them and do not repeat again.

**Audience:**

Yes

**Broader Impact Concerns:**

None.

**Claims And Evidence:**

Yes

**Requested Changes:**

The current version suffers from some format issues.
1. Some citation formats are incorrect. I suggest the authors to figure out the differences among \cite, \citet, and \citep. For example, Line 2 in the introduction should be "(Rodrigues et.al, 2019; Xu & Wunsch, 2005)."
2. For the footnote, it should come after the period without a space.

**Strengths And Weaknesses:**

Strengths
1. The reproducibility verifies the contributions of fairness attack and consensus fair clustering in the original paper.
2. The extended work is a plus to make the original method more stronger.
3. In general, this paper is in good shape, which is clear and easy to follow. I do not have major concerns.

Weaknesses
1. For the extended work, it would be better to provide their motivations.

---

> ### Author Response · Authors · 2024-04-01
> **Reply to Reviewer XiUa**
>
> Dear Reviewer XiUa,
>
> Thank you for your feedback. We have revised our paper accordingly and hope the changes meet your expectations:
>
> We corrected citation formats as advised, ensuring proper use of \cite, \citet, and \citep throughout the manuscript.
> Footnotes now correctly follow the punctuation, without any space.
> Each extended work section (4.2.1, 4.2.2, 4.3) now includes a motivation paragraph to provide clarity on our objectives.
>
> Please let us know if there are any further modifications you believe are necessary or if our changes align with your expectations.
>
> Thank you again for your constructive feedback.
>
> Best regards,
> The Authors

---

> > ### Comment · Reviewer_XiUa · 2024-04-01
> > **Response to Authors**
> >
> > Dear Authors,
> >
> > Thanks for these revisions. I have no further comments.
> >
> > Kind Regards,
> >
> > Reviewer XiUa

---

### Review · Reviewer_dHNL · 2024-03-19

**Summary Of Contributions:**

The paper revisits the work "Robust fair clustering: A novel fairness attack and defense framework" by Chhabra et al. (2023). Specifically, focusing on the evaluation of the proposed Consensus Fair Clustering (CFC) model, the paper conducts a reproducibility study on Chhabra et al. (2023), cleans up the original code repository, considers additional metrics (for clustering), data sets, and attack methods, and also conducts an ablation study. The original three claims in Chhabra et al. (2023) are mostly reproduced/validated, and the overall evaluation of CFC model in this work is more comprehensive than that in the original paper.

**Audience:**

Yes

**Broader Impact Concerns:**

I do not foresee potential ethical concerns of this reproducibility study.

**Claims And Evidence:**

Yes

**Requested Changes:**

Minor adjustment:

- provide some technical detail about the additional methods considered in Section 3.2 (for the purpose of clearly presenting the importance of considering these approaches, e.g., how they relate to the CFC model and the claims by Chhabra et al., 2023)

**Strengths And Weaknesses:**

Overall, the paper is relatively easy to follow. The paper revisits the major claims in Chhabra et al. (2023) and (mostly) validates them with reproducibility studies. The extended experiments are clearly documented and presented, and the overall evaluation of the CFC model appears to be more comprehensive than that when CFC was proposed in Chhabra et al. (2023).

The paper can be further improved by clearly stating why certain additional metrics and attack methods are considered (i.e., why these choices) and providing more mathematical details in Section 3.2 if space permits (i.e., how things work, going beyond narrative descriptions). This way, the importance of additional experiments considered in this reproducibility study can be better positioned and appreciated.

---

> ### Author Response · Authors · 2024-04-01
> **Reply to Reviewer dHNL**
>
> Dear Reviewer dHNL,
>
> Thank you for your feedback. We have revised our paper accordingly and hope the changes meet your expectations:
>
> We have included a paragraph in Section 3.2 highlighting how the described models relate to the CFC model.
> Furthermore, we have included a motivation paragraph for each extended work section (4.2.1, 4.2.2, 4.3) to provide clarity on our objectives.
>
> Please let us know if there are any further modifications you believe are necessary or if our changes align with your expectations.
>
> Thank you again for your constructive feedback.
>
> Best regards, The Authors

---

### Review · Reviewer_oUwk · 2024-03-19

**Summary Of Contributions:**

The paper conducts a reproducibility study on "Robust Fair Clustering: A Novel Fairness Attack and Defense Framework," focusing on the robustness of fair clustering algorithms against adversarial attacks. It examines the vulnerability of three fair clustering models (Fair K-Center, Fair Spectral Clustering, and Scalable Fairlet Decomposition) to such attacks, confirming their susceptibility. Additionally, the study evaluates the resilience of the Consensus Fair Clustering (CFC) model against these fairness attacks. Beyond replicating the original study's findings, the paper extends the work by enhancing the experimentation codebase, integrating additional metrics and datasets for a deeper analysis, implementing new attack methods, and conducting an ablation study on hyperparameters within the CFC model. The authors partially confirm the original study's claim that adversarial attacks can degrade fairness performance by altering protected group memberships, fully reproduce the claim regarding the existing models' lack of robustness, and affirm CFC's high resilience to fairness attacks.

**Audience:**

Yes

**Broader Impact Concerns:**

None.

**Claims And Evidence:**

Yes

**Requested Changes:**

The partial reproduction of the first claim requires further clarification. Specifically, a detailed analysis explaining the possible reasons behind the partial reproduction and how it might affect the overall conclusions of the study would be critical.

**Strengths And Weaknesses:**

Strengths:
The study successfully reproduces the key findings of the original paper and extends the analysis with additional datasets, metrics, and attack methods, contributing to a deeper understanding of the robustness of fair clustering algorithms.

The introduction and evaluation of new attack strategies against fair clustering algorithms offer insights into potential vulnerabilities and the effectiveness of current defense mechanisms.

Weaknesses:
I am primarily concerned about whether a study focused mostly on replicating the findings of a single paper would attract enough interest from the research community. This paper does not offer an extensive survey of the literature on fair clustering or fairness attacks. Although it brings in new datasets and metrics, the assessment largely relies on the baselines established by the reviewed paper. As a result, the study does not provide readers with a comprehensive overview of the current state of research in fairness attacks and robust fair machine learning.

---

> ### Author Response · Authors · 2024-04-01
> **Reply to Reviewer oUwk**
>
> Dear Reviewer oUwk,
>
> Thank you for your valuable feedback. We have revised our paper accordingly and hope the changes meet your expectations:
>
> - Detailed Analysis for Claim 1: We've expanded our discussion with an in-depth analysis explaining the partial reproduction of Claim 1, including possible reasons and its impact on our study's conclusions.
> - Broader Impact: A new paragraph at the end of our discussion now highlights the broader significance of our study within the ML literature.
> - Objectives of Extended Work Sections: We've included motivation paragraphs in Sections 4.2.1, 4.2.2, and 4.3 to clarify the purpose and contribution of our extended work.
>
> Please let us know if there are any further modifications you believe are necessary or if our changes align with your expectations.
>
> Thank you again for your constructive feedback.
>
> Best regards,
>
> The Authors

---

> > ### Comment · Reviewer_oUwk · 2024-04-07
> >
> > Thanks for your response. I have no further comments.

---

### Decision · Action_Editor_AqK7 · 2024-04-09

**Recommendation:** Accept as is

**Comment:**

The paper presents a reproducibility study of "Robust Fair Clustering: A Novel Fairness Attack and Defense Framework" and extends the analysis to provide additional insights into the robustness of fair clustering algorithms. Specifically, the authors partially confirm the original study's claim that adversarial attacks can degrade fairness performance by altering protected group memberships, fully reproduce the claim regarding the existing models' lack of robustness, and affirm the high resilience of the Consensus Fair Clustering (CFC) model to fairness attacks. Additionally, this paper extends the work by enhancing the experimentation codebase, integrating additional metrics and datasets, implementing new attack methods, and conducting an ablation study on hyperparameters within the CFC model.

During the rebuttal phase, the authors addressed concerns including the depth of analysis by Reviewer oUwk; additional motivation and technical details about the methods used by Reviewer dHNL and Reviewer XiUa; footnote and citation format by Reviewer XiUa.

After the rebuttal, all reviewers acknowledged that there were no further concerns and expressed their willingness to accept this paper. Therefore, we recommend acceptance.

**Audience:**

This paper can be interesting to researchers in fair clustering and adversarial attacks.

**Claims And Evidence:**

Most of the claims are clear, convincing, and supported by evidence.

---

> ### Author Response · Authors · 2024-04-10
> **Re: Decision on Submission**
>
> Dear Action Editor AqK7 and Reviewers,
>
> Thank you for accepting our manuscript and for your constructive comments throughout the review process. We appreciate the opportunity to contribute to the field and will follow any further instructions needed for publication.
>
> Best regards,
>
> The authors